# The Martensitic Transformation and Mechanical Properties of Ti6Al4V Prepared via Selective Laser Melting

**DOI:** 10.3390/ma12020321

**Published:** 2019-01-21

**Authors:** Junjie He, Duosheng Li, Wugui Jiang, Liming Ke, Guohua Qin, Yin Ye, Qinghua Qin, Dachuang Qiu

**Affiliations:** 1School of Materials Science and Engineering, Nanchang Hangkong University, Nanchang 330063, China; 233hjj@gmail.com (J.H.); 13627007624@163.com (Y.Y.); Dachuangq@gmail.com (D.Q.); 2School of aeronautical manufacturing and engineering, Nanchang Hangkong University, Nanchang 330063, China; jiangwugui@nchu.edu.cn (W.J.); limingke@nchu.edu.cn (L.K.); qghwzx@126.com (G.Q.); 3Research School of Engineering, Australian National University, Acton, ACT 2601, Australia; qinghua.qin@anu.edu.au

**Keywords:** selective laser melting, Ti6Al4V alloy, martensitic transformation, texture evolution, mechanical properties

## Abstract

This article investigated the microstructure of Ti6Al4V that was fabricated via selective laser melting; specifically, the mechanism of martensitic transformation and relationship among parent β phase, martensite (α’) and newly generated β phase that formed in the present experiments were elucidated. The primary X-ray diffraction (XRD), transmission electron microscopy (TEM) and tensile test were combined to discuss the relationship between α’, β phase and mechanical properties. The average width of each coarse β columnar grain is 80–160 μm, which is in agreement with the width of a laser scanning track. The result revealed a further relationship between β columnar grain and laser scanning track. Additionally, the high dislocation density, stacking faults and the typical ($10\bar{1}1$) twinning were identified in the as-built sample. The twinning was filled with many dislocation lines that exhibited apparent slip systems of climbing and cross-slip. Moreover, the α + β phase with fine dislocation lines and residual twinning were observed in the stress relieving sample. Furthermore, both as-built and stress-relieved samples had a better homogeneous density and finer grains in the center area than in the edge area, displaying good mechanical properties by Feature-Scan. The α’ phase resulted in the improvement of tensile strength and hardness and decrease of plasticity, while the newly generated β phase resulted in a decrease of strength and enhancement of plasticity. The poor plasticity was ascribed to the different print mode, remained support structures and large thermal stresses.

## 1. Introduction

As opposed to traditional subtractive manufacturing, selective laser melting (SLM) is a layer-by-layer overlapping technology that was used to create three-dimensional (3D) components from 3D model data by using laser as the input heat source. SLM has attracted much attention over the past few years for its immanent advantages, such as high material utilization, short work cycle time and the ignorance of the geometric shape during processing. Furthermore, SLM is an environmentally friendly and advanced manufacturing technology that is widely used in processing titanium, Co-Cr, aluminum and stainless alloys in different fields [1,2,3,4,5,6,7].

As a kind of α + β dual-phase titanium alloy, Ti6Al4V has been widely used in national defense, automobile and medicine due to its excellent performance including good anticorrosive, low density, superior weldability and high specific strength. However, it is difficult to machine Ti6Al4V alloy via conventional measures for its high melting point, frangibility to oxidation, large deformation resistance and poor cutting ability. Those characteristics lead to high production costs [8]. In addition, the low surface roughness and poor plasticity of Ti6Al4V alloys that were fabricated by SLM are not ideal; these have caused the property of SLM parts to be of lower quality when compared to traditional manufactured parts.

Recently, many experiments have focused on the mechanical properties and microstructures of the Ti6Al4V alloy. For instance, the morphology, utilization and particle size of additive manufacturing powder has been reviewed [9,10]. The as-built sample and its support structures were investigated to explore the performance of the region in sample and support structures, respectively [11,12]. The surface roughness of as-received and laser polished samples of Ti6Al4V and TC11 alloys were also compared, respectively [13]. Moreover, researches have reported on the relationship between the sample quality and the process parameters as laser power, exposure time, point distance, particle size, layer thickness and hatching distance during SLM [14,15,16]. The effect of six different kinds of scan patterns on microstructure and mechanical properties has been investigated [17]. The microstructures and mechanical properties of different alloys manufactured by various additive manufacturing (AM) technologies were introduced, respectively [3,18]. While the typical hexagonal close-packed (HCP) diffraction spots were presented and that α’ martensite was the only structure in SLM samples [19].

Nowadays, many studies have focused on the mechanical properties and microstructures of Ti6Al4V alloys, and different heat treatments or different processing technologies have been designed to promote the comprehensive property of Ti6Al4V. Little attention has been paid to the origin of martensitic transformation mechanism during SLM between the parent β phase and the newly generated α’ phase in-depth, owing to its complexity. This study focused on the martensite that is generated from Ti6Al4V by SLM, which is distinct from the traditional subtractive methodologies. The deformation mechanism related to the β phase and α’ phase and the influence of microstructure on macroscopic mechanical properties were also elucidated. Furthermore, the commercial SLM printer was used to build samples with different print mode.

## 2. Material Preparation and Methods

In the present work, the SLM system (Space Traveler Tr150, Profeta Intelligent Technology, Nanjing, China) was used to fabricate the rectangular and cube parts as shown in Figure 1. The optimal process parameters are listed in Table 1.

The plasma atomized Ti6Al4V-ELI powder with an average diameter of 34 μm was used in the present experiments. Figure 2a shows the powder morphology under scanning electron microscope (SEM, FEI Quanta 200, Hillsboro, OR, USA). Figure 2b shows the powder size and composition of the Ti6Al4V powder that were obtained by laser-based particle size analyzer and energy dispersive spectrometer (EDS) respectively. The high sphericity and smooth-faced powder has a narrow size distribution between 10 μm and 50 μm in which the particle diameter of 37 μm occupies highest proportion. Table 2 displays the detail chemical composition of the experimental Ti6Al4V powder.

The laying powder device spread the powder across the building substrate, and then, a laser beam scanned the uniform powder. The laser scanned the area of each layer with parallel scan vectors and the border of each layer was then scanned twice (interior and exterior contour). The scan direction initially fused at a 45° angle to the horizontal line and then rotated by 90° between each new layer as shown in Figure 3. The laser had a power of 135 W; a scan speed of 800 mm/s was applied to the center area of the parts, which were in variance from the border where the laser power was lower. A reduced scan speed resolved the problem of energy accumulation. In this study, the edge area of sample was optimized to be easily removed by post-machining once the print process was completed; however, due to the surface roughness of SLM, certain parts did not satisfy the requirement of the surface roughness national standard. This kind of design pattern is different from the related report in which the print parameters were identical in both the center area and edge area [19]. Additionally, the laser parameters of supporting structure were designed separately on the basis of the principle of easily detachment from the specimens. The height of supports was 2 mm with 0.2 mm-diameter of each fine column. The base plate had been preheated to decrease the occurrence of thermal stress. All the samples were built in the argon atmosphere. Moreover, the porosity of the shaped sample that was measured ranged from 0.33% to 0.69% by using Archimedes method.

The hardness of as-built and stress relieving samples was determined with a Vicker’s indentation test machine (HX-1000TM, Shanghai optical instrument factory, Shanghai, China) that performed with a load of 500 gf for 5 s at each point. Moreover, the Feature-Scan system (UTF Scan-surface and holes, Nanchang Huali Oil Testing Company, Nanchang, China) was also carried out to detect the defects in as-built and stress relieving samples before the tensile test. The phase composition of as-built and stress relieving samples was analyzed using the software Image-Pro Plus 6.0 (Media Cybernetics, Maryland Co, Rockville, MD, USA), wherein three optical microscopy (OM) (DM1500, Shenzhen mass photoelectric company, Shenzhen, China) images at different position were measured respectively.

The rectangle specimens were fabricated directly into a dog bone shape in three orientations, which were built with a gauge length, width and thickness of 12 mm, 3 mm and 3 mm, respectively. The tensile test was conducted at room temperature with a strain rate of 2 mm/min. Heat treatment was designed at 1003 K for 2 h in N_2_ gas conditions. Once the heat preservation process was completed, the specimens were furnace cooled [20,21,22,23,24,25]. Both as-built and stress relieving specimens were then polished in the support surface and tested by Instron 8827 tensile system (Instron Corporation, Norwood, MA, USA).

The microstructure and morphology of the cube samples were analyzed using X-ray diffraction (XRD, Bruker D8 Advance, Billerica, MA, USA), optical microscopy (OM), scanning electron microscopy (SEM), and transmission electron microscopy (TEM, Talos F200X, FEI Company, Hillsboro, OR, USA). The cube samples for OM were mechanical polished and then etched in kroll solutions consisting of 2 vol.% HF, 5 vol.% HNO_3_ and 43 vol.% H_2_O [19]. The prepared Ti6Al4V flakes for TEM were polished via twin-jet electropolishing device in mixed solution of 5 vol.% perchloric acid and 95 vol.% anhydrous ethanol at −30 °C.

## 3. Results and Discussion

### 3.1. Microstructure

#### 3.1.1. XRD Analysis

At the beginning of the SLM process, the dual-phase structure at the room temperature transforms rapidly which result in the presence of martensitic α’ phase (HCP). The β phase did not have enough time to precipitate α phase in such a large undercooling (even more than one thousand degrees per second); in other words, the composition of prior β phase was almost unchanged, although the crystal lattice did change. As a result, the α phase and the substitutional supersaturated solid solution α’ phase generated by this lattice transformation [26].

Commonly, the 100% martensite is impossible to obtain under this large undercooling [26]; thus, the XRD pattern in Figure 4 displays the close-packed hexagonal structure of α/α’ martensitic peaks that were similar to other published reports in as-built samples [19,27]. A little deviation of the α/α’ phase peak has been observed in comparison with the standard diffraction peak of α-Ti after X-ray diffraction analysis. The phenomenon may derive from the solid strengthening and the other factor is the Ti6AL4V crystal structure distortion. The smaller atomic radius of the V elements dissolved in the Ti lattice, which has a larger atomic radius that led to a crystal structure distortion and then the migration of the diffraction peaks had taken place.

Figure 4 shows that the intensity of peaks of the as-built sample were higher than the stress-relieving one, which indicates that the stress relieving samples possessed a finer α/α’ phase. The result was attributed to a part of the α’ phase transforms into the α phase and β phase, and the other reason is the α phase precipitates the β phase during stress relieving. This can be proven with the stress-relieving XRD pattern. Here, a new β phase was observed because of the decomposition of the α’ phase or the precipitation from the α phase. The details were analyzed in the following research.

On the other hand, the decrease of strength and hardness in stress relieving samples were also an important signal of the α’ phase transformation. The acicular martensite formed by SLM was a substitutional supersaturated solid solution (α’), which had a positive enhancement in strength and hardness for the whole structure. However, the acicular martensite easily decomposed after stress relieving for its instability. The above two factors resulted in the increase in strength of Ti6Al4V; therefore, the hardness improved a little. Moreover, the driving force for martensitic transformation of Ti alloy was low (approximately −25 J/mol). The thermal hysteresis that was used to characterize the difference between the initial temperature of transformation and start temperature of martensite reverse transformation was rather small. All these phenomena contributed to the precipitation of β phase during the heat treatment.

#### 3.1.2. Microstructural Mechanism

In this research, the martensite transformation was a first-order phase transition; all the parent β phase atoms transformed to an α/α’ phase as a lattice reconstruction without diffusion. Thus, the first-order derivative of chemical potential were not equal according to the Equations (1)–(4):(1)$${\left(\frac{\partial \mu }{\partial P}\right)}_{T}\neq {\left(\frac{\partial \mu^{\beta }}{\partial P}\right)}_{T},$$
(2)$${\left(\frac{\partial \mu }{\partial P}\right)}_{T}=V,\;V^{\alpha }\neq V^{\beta }.$$
(3)$${\left(\frac{\partial \mu^{\alpha }}{\partial T}\right)}_{P}\neq {\left(\frac{\partial \mu^{\beta }}{\partial T}\right)}_{P},$$
(4)$${\left(\frac{\partial \mu }{\partial T}\right)}_{P}=-S,\;S^{\alpha }\neq S^{\beta }.$$
where *μ* is the chemical potential, *P* is pressure and *T* is temperature. The formulas indicate the changes of volume and entropy between the β phase and α/α’ phase exist in a first-order phase transition; in other words, volume change and the release (or absorption) of latent heat happened during SLM. The chemical compositions of the new phase and parent phase were identical, while the crystal structure and specific volume were quite different. For these reasons, the huge thermal stress and distortional energy were produced within the as-built alloys [28,29], which brought high dislocation density, stacking faults and twin substructure.

The typical acicular martensitic phase that originated from the prior β boundaries and filled in the β columnar grains are visible in Figure 5a. Each acicular martensite with an average width of 2 ± 0.5 μm, grew spontaneously along the preferred orientation dictated according to the special Burgers relation between the β phase and α/α’ phase during the deposition process [30]. Additionally, the average width of β columnar grains was measured at a range of 120 ± 20 μm, which was coincidentally equal to the width of scan track. The similar phenomenon were discussed by Simonelli, M. et al. [31], in which the average width of β columnar grain in Ti6Al4V was 210 ± 50 μm. Curiously, the ratio of the scan track width to the columnar grain width was approximately unitary, owing to the large undercooling once the laser scanned; the content within the β phase transformed to an α’ texture by crystal reconstruction instead of the precipitated α phase and the fine acicular martensite formed by preferential growth. Eventually, the morphology of β columnar grains with many acicular martensite emerged.

The size of martensite after stress relieving was measured with an average width of 0.4 ± 0.1 μm. These acicular martensite became shallower than that in the Figure 5a. The fine α phase and β phase can be observed from Figure 5b, indicating the decomposition of α’ phase. It is generally known that elastic strain energy improved with the martensite growth, and conversely, the excessive energy block the growth of martensite. Comparing with as-built image, the martensite in stress-relieved image is smaller, indicating the reduction of elastic strain energy.

Furthermore, the phase composition of both samples in Table 3 demonstrated that the α’ phase decreased after stress relieving, while the proportion of α’ phase reduced from 28.50 to 9.90. However, the newly generated β phase rose from 0 to 10.55%. Since the annealing temperature was far from reaching the β transus temperature, there was either no β phase or the number of β phase was too small to be detected by the used methods in as-built samples. The newly generated β phase in the XRD pattern after stress relieving was attributed to two factors. The first reason was the instability of the α’ phase. The non-equilibrium α’ phase was easy to break down into an α+β phase as a metastable phase, which ascribe to its own characterization [21,27]. Moreover, the α’ martensite of Ti6Al4V in SLM could be considered as thermal elastic martensitic owing to its low thermal hysteresis, which gives an appropriate explanation about the decomposition of α’ phase. Secondly, a 2 h furnace cooling condition during stress relieving was regarded as an aging treatment, which resulted in the appearance of the β phase.

The acoustic wave detection was applied by a Feature-Scan system to study the microstructure homogeneity and density distribution of the samples in as-built and stress relieving conditions. The echo amplitude diagram is shown in Figure 5c. The wave intensity in red area is higher than in the green area. This phenomenon appeared because of the acoustic attenuation. Once the sound wave propagated in the boundaries (green area), the waves greatly attenuated due to the slag and blowholes. On the other hand, in the center area (red area), there were relatively few defects with fine grains and excellent density. Thus, the intensity of the bounced sound in center area was higher than that of the boundaries. As in the article mentioned before, the boundaries of parts were devised in different process parameters to remove easily to reach the surface roughness standard, while the content in the center of the sample were not affected. Hence, the result reflected the different homogeneous of microstructure or density distribution. In addition, the stress-relieving sample exhibited a better homogeneous density and finer grains in the center area than the as-built sample, indicating the diminish of void ratio, grain refinement and the reduction of residual stress.

The α’ acicular martensite transformation of Ti6Al4V during SLM has been introduced in detail in the preceding paragraphs. As it turns out, the newly generated metastable α’ phase with close-packed hexagonal structure originated from the parent β phase, which is a body-centered cubic lattice. Generally, a relationship of coherent or semi-coherent interface always existed between the new phase and the prior phase in martensitic transformation. The prior β phase possessed a dominant <100> texture, which is a well know grain growth direction. The newly generated α’ phase had a typically crystallographic orientation relationship with the prior β phase (0001)_hcp_∥(110)_bcc_, <$11\bar{2}0$>_hcp_∥<111>_bcc_ [21,32,33,34].

The apparently higher dislocation density, fine twin substructure and stacking faults are displayed in as-built samples in Figure 6. Since the inner regions subjected uneven heating and cooling cyclically, the large thermal stress occurred during the SLM deposition process, and it was no doubt why so many higher dislocation density and twins were found in the as-built sample. The high amount of twins and the high dislocation density indicated that the high inner stresses certainly existed, which resulted in the improvement of yield strength (YS) and ultimate tensile strength (UTS). Furthermore, based on the mechanism of new martensite transformation [35], the formation of newly emerged α’ martensite was preferentially growth through the direction of lower strain energy and as a result, these meta-structures occurred in order to adjust the strain energy during the process of lattice modification.

The twins in the present experiment were regarded to be transformation twins. The lattice of the twins was modified instead of emerging deformation twins that ascribed to applied stress. The width of twins was in a range distribution from 30–100 nm. The twin planes, for instance, {$10\bar{1}2$} and {$11\bar{2}1$}, were observed as extension twins in other reports [36]. In this study, the typical {$10\bar{1}1$} twinning plane were calibrated as the only type of twins by selective area electron diffraction (SAED) data as shown in Figure 6e,f. These types of twins in HCP materials are defined as contraction twins. From Figure 6b,c, it is quite evident that each bamboo-like twin lamellar are glutted with many dislocation lines, which indicates that the slip system of climbing and cross-slip occurred in the deposition process. Beyond that, the average width of neighboring stacking faults was gauged to be about 70 nm wide.

The stress relieving sample in Figure 6g,h shows a disorderly dual phase structure of α + β phase during TEM, which was anticipated due to the martensite decomposition in stress relieving condition differs from the 3D printing process. The reason is that the sample was in a relatively uniform temperature during annealing treatment, while the heat distribution of SLM process was inhomogeneous. Additionally, the residual twins were found in stress-relieved samples, as marked in Figure 6h. Different from the as-built images, the higher density of dislocations did not exist; instead, the fine dislocation lines were discovered in Figure 6i. Thus, the effect of stress relieving was obvious for the high dislocation density, and stacking faults disappeared, twin structure decreased and β phase generated. Such a result differed from the literature [21], where the high dislocation density and stacking faults can also be detected and no β phase in stress-relieved samples. On a macroscopic scale, owing to the reduction of internal stresses and the transformation of α’→α + β, the tensile strength and hardness of stress-relieved samples were decreased and the plasticity was increased.

### 3.2. Mechanical Properties

#### 3.2.1. Tensile Properties

The performance of three different building orientation samples have been discussed in detail [20]; thus, two batches with six edge oriented samples were selected to study the mechanical properties in as-built and stress-relieved parts by tensile experiment. The stress-strain curve was shown in Figure 7. Additionally, the elastic modulus (E), YS, UTS, fracture stress (FS), fracture elongation (ε fracture) and hardness (H) were also analyzed, as shown in Table 4. The hardness (H) of both samples were in agreement with other reports [20,37].

The UTS of the as-built sample raised with 9.3% compared to the stress-relieving sample. The elastic modulus of stress-relieving sample was the same as that of the as-built sample. In the as-built sample, the yield strain was less than 2% and the fracture strain was tested to be less than 3% [12]. However, the plasticity improved through stress relieving and the average fracture strain rose from 1.9% to 3.6%. The hardness of stress relieving samples was lower than the as-built sample in the upper surface.

Compared to as-built sample, it was evident that the properties of the stress relieving sample changed. The presence of obvious plastic deformation in stress relieving line was distinct with the as-built curve which almost no plasticity deformation observed. As mentioned in Figure 4, the existence of α’ martensitic phase had a prominent improvement in strength and hardness, while it had a negative influence in plasticity, which contributed to the higher strength and hardness before being stress-relieved and better plasticity once the α’→α + β phase transformation took place. As noted in Figure 5a,b, the martensite size of stress-relieved sample was smaller than the as-built one, indicating that the elastic strain energy reduced. Thus, the YS and UTS decreased and plasticity improved.

As a β-phase stabilizing element, the V element was a main ingredient in the α + β Ti alloy that caused the morphology of acicular martensite differ from other near α titanium alloys of lower V element and plate martensite. In Ti6Al4V, the V element adopted the mode of displacement to dissolve in β-Ti as a β isomorphous element; thus, the crystal distortion was small and resulted in the low internal stresses and distortional energy. As a result, the strength and hardness of α’ martensite were a little higher than the α phase. However, the strength and hardness of this kind of α’ martensite did not significantly improve compared with martensite in steel.

In addition, the elastic modulus of stress relieving sample did not vary much compared with as-built one, indicating the total number of the α’/α phase without significant change because of the anisotropic α’/α structure had a marked influence on elastic modulus of samples. Nevertheless, the β texture had no influence on elastic modulus. The details were discussed in the related report [20].

Though the UTS/YS is higher than reported in many previous researches, the fracture strain of both as-built and stress relieving samples was too low compared with other related literature [21]. The three reason are summarized here. Firstly, as mentioned in Section 2, unlike other literature where the print parameters are same in total sample, the present experiment had two sets of parameters applied in the center area and edge area, respectively. The brittle edge area was designed to be easily removed by post-machining for commercial application, which was probably the main reason that resulted in the samples yielded early, and premature fracture during tensile test. Secondly, though the support structure area was mechanically polished, pits and bumps still remained in such a brittle sample surface, which may accelerate the fracture process. Additionally, as shown in Figure 6 as-built images, there were many high densities of dislocation and twins, indicating the larger thermal stresses typically occurred during SLM, which led to the error to come out in the tensile test due to the samples being a little curled.

#### 3.2.2. Tensile Fracture Mechanisms

The transgranular fracture surface of the edge orientation in both the as-built and stress-relieved samples is shown in Figure 8. The microstructure exhibited a typical mixed mode of brittle and ductile fracture that were characterized as the observable cleavage facets, microvoids and shallow dimples [19,20].

Since the fracture strain was less than 3%, the dimple was non-existent. As shown in Figure 8a,c,d, the average diameter of these obvious pores was 38 ± 10 μm in as-built samples and the pores’ size in stress relieving samples were reduced slightly: about 32 ± 8 μm wide. Particularly, these lake ripples around the pores in Figure 8c had been considered as reinforcements. However, the EDS image indicated that the compositions of this rippled structure were coincided with elsewhere in the samples. It is generally known that the excess laser power or lower scan speed can cause over-heating (OH), and low heat input or faster scan speed may result in the incomplete melting (IM). Based on this, the multi-layer pores were ascribed to the OH or IM during the SLM process, and each ripple represented one deposition layer. Once the OH or IM happened, the pore of previous layer could not be coated by the following powder layers. Hence, the fused powders of following layers exhibited the sign of ripples.

Moreover, a large number of fine grain structures were observed in stress-relieved samples that were marked in Figure 8e, which corroborated that the grains were refined after stress relieving. According to the Hall-Petch strengthening mechanism, smaller grain size will provide more grain boundaries, which can impede the movement of dislocation. Then, the hardness, YS and UTS improved. Nevertheless, the strength of the sample decreased simultaneously for a part of the unstable α’ phase transformed into an α + β phase, the number of twins significantly decreased and the high dislocation density disappeared after an annealing treatment. Finally, the plasticity of the stress-relieved samples increased, and the YS and UTS were reduced. The result coincided considerably with Figure 7 and the tensile data in Table 4. The unmelted powder particles near the pores are marked in Figure 8f, indicating the IM phenomenon generated [38].

## 4. Conclusions

The microstructure and mechanical performance of both as-built and stress-relieved Ti6Al4V alloys by SLM technology were investigated in detail, and the main conclusions are as follows:(1)After stress relieving, the β phase was observed in XRD pattern, OM metallograph and TEM images, and the composition of α’ phase reduced, indicating that the α’ phase decomposed during annealing treatment.(2)The width of the coarse β phase may have a further relationship with the width of laser scan track.(3)In as-built samples, the high dislocation density, twinning and stacking faults were detected, and the typical ($10\bar{1}1$) twinning plane were marked where the clearly dislocation lines were discovered. In stress-relieved samples, a fully disordered α + β phase filled in the TEM images, and there were neither high density of dislocation nor stacking faults. Only fine dislocation lines and a little twinning remained.(4)α’ phase resulted in an increase in tensile strength and hardness and a decrease in plasticity. The poor plasticity was ascribed to the print mode, remained support structures and large thermal stresses.

## Figures and Tables

**Figure 1 materials-12-00321-f001:**
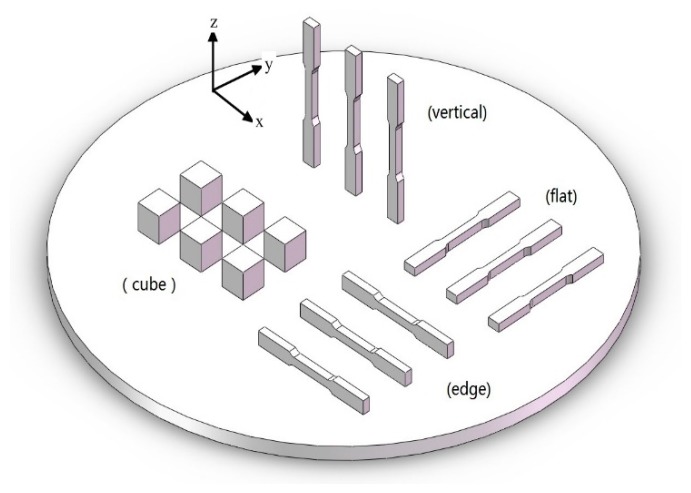
Computer-aided design (CAD) models of cubes and three orthogonal orientations (flat, edge and vertical) tensile samples on the stainless steel substrate.

**Figure 2 materials-12-00321-f002:**
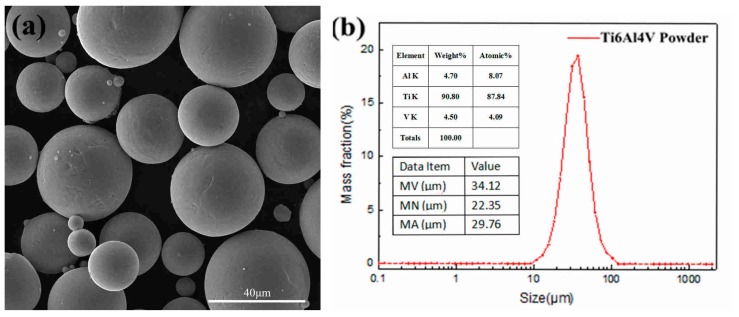
(**a**) The morphology of Ti6Al4V powder; (**b**) particle size and distribution.

**Figure 3 materials-12-00321-f003:**
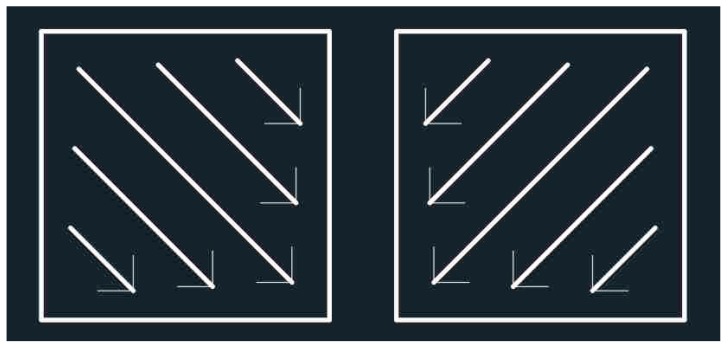
The scan pattern of selective laser melting (SLM).

**Figure 4 materials-12-00321-f004:**
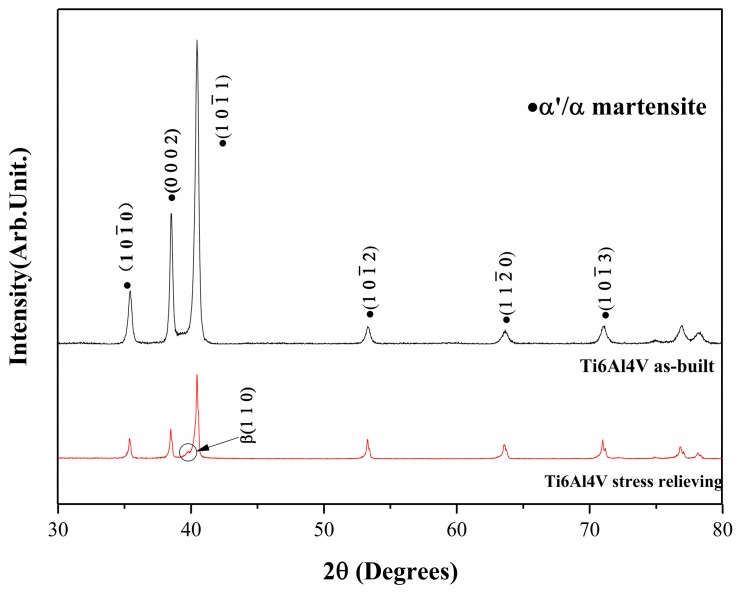
XRD patterns of as-built and stress relieving samples.

**Figure 5 materials-12-00321-f005:**
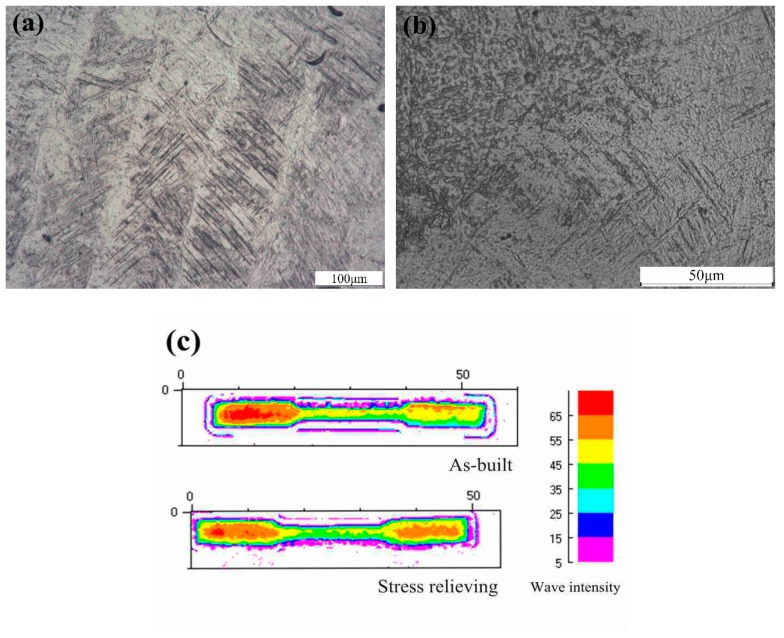
Optical microscopy in top surface: (**a**) the as-built sample; (**b**) middle portion of the stress relieving sample and (**c**) the echo amplitude diagram.

**Figure 6 materials-12-00321-f006:**
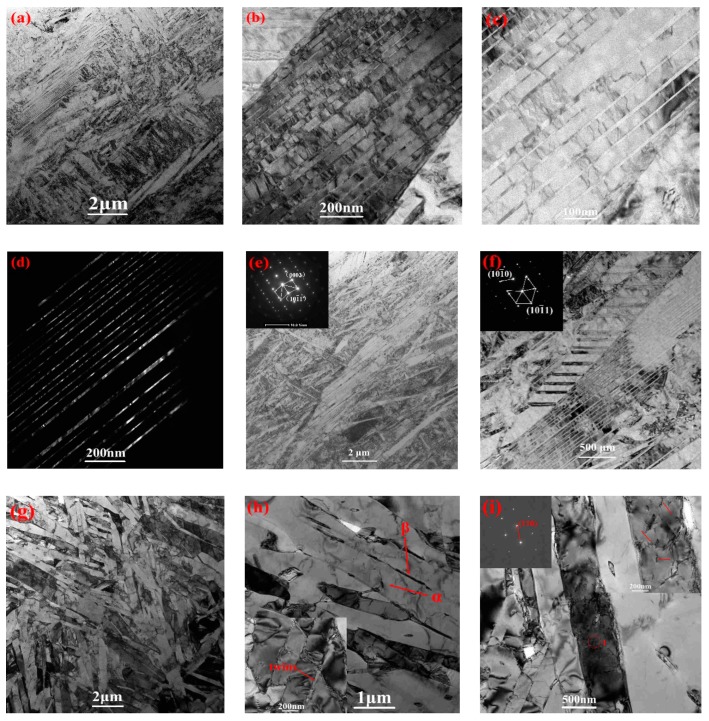
Transmission electron microscopy (TEM) of as-built cube samples: bright field (**a**–**c**); dark field (**d**); twinning and diffraction pattern (**e**,**f**) and stress relieving samples of bright field (**g**–**i**).

**Figure 7 materials-12-00321-f007:**
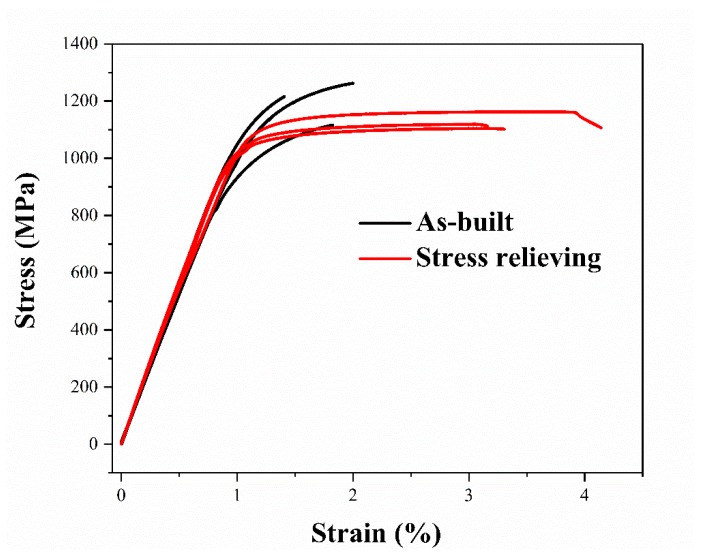
Stress strain curves of as-built and stress relieving Ti6Al4V parts.

**Figure 8 materials-12-00321-f008:**
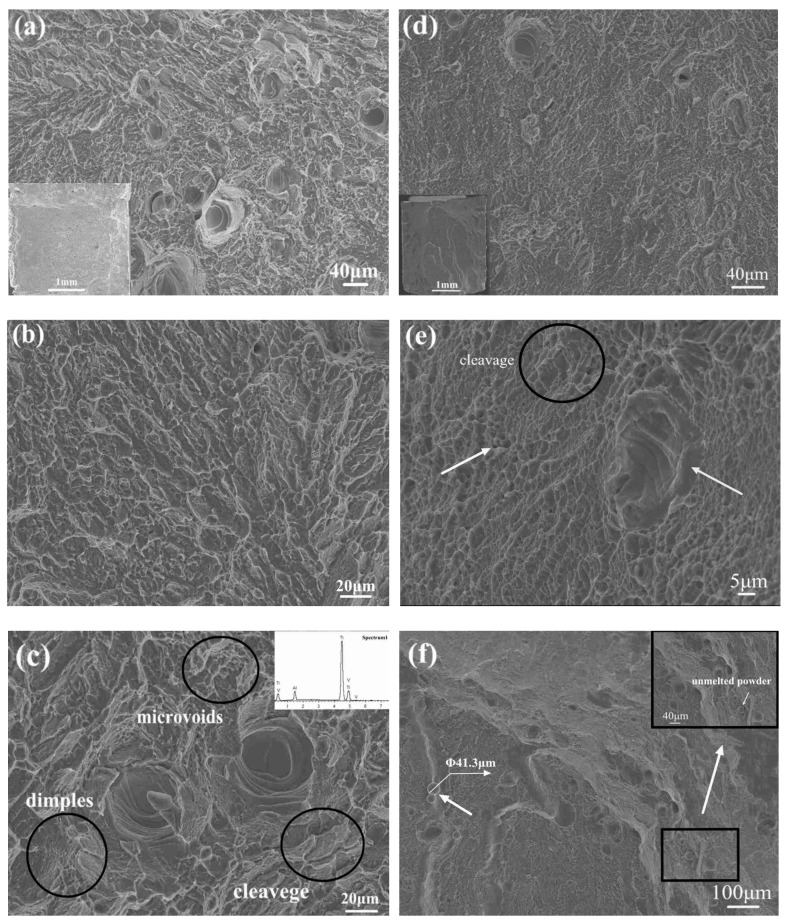
Ti6Al4V fracture surface of the tensile specimens (edge orientation): as-built parts (**a**–**c**); stress-relieved parts (**d**–**f**).

**Table 1 materials-12-00321-t001:** Process parameters of preparing Ti6Al4V specimens.

Process Parameters	Values
Laser power (W)	135
Layer thickness (μm)	30
Exposure time (μs)	400
Scan speed (mm/s)	800
Laser spot size (μm)	52

**Table 2 materials-12-00321-t002:** Chemical composition of Ti6Al4V powder (wt %).

Element	Ti	Al	V	O
Ti6Al4V	89.84	6.25	3.90	<0.1

**Table 3 materials-12-00321-t003:** The phase composition of as-built and stress relieving samples.

Samples	Phase		
	α’/%	α/%	β/%
As-built	28.50	71.50	0
Stress relieving	9.90	79.55	10.55

**Table 4 materials-12-00321-t004:** Tensile properties of SLM Ti6Al4V samples.

Parameters	Elastic Modulus (GPa)	Yield Strength (MPa)	Ultimate Tensile Strength (MPa)	Fracture Stress (MPa)	Fracture Elongation (%)	Hardness (HV)
As-built	107 ± 4	1142 ± 17	1235 ± 37	1235 ± 37	1.3 ± 0.5	395 ± 21
Stress relieving	114 ± 2	1057 ± 25	1130 ± 30	1129 ± 30	2.8 ± 0.4	390 ± 18

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
