# Peer review of "The Martensitic Transformation and Mechanical Properties of Ti6Al4V Prepared via Selective Laser Melting"

_materials, 2019, doi:10.3390/ma12020321_

Round 1
Reviewer 1 Report
This paper describes the experimental results of laser melting/thermal-modification of Ti6Al4V and evalutions of its proterties. In terms of the laser-additive manufacturing, the results obtained in the manuscript seems to be useful for industry, in addition to a hard-to-work material of Ti6Al4V. Thus, this paper is acceptable for the publication to this journal, Materials.
Author Response
Thanks for the referee’s kind comments, we appreciate your affirmation on our work.
Reviewer 2 Report
The article describes the mechanism of martensitic transformation that occurs during SLM process of Ti6Al4V alloy and its influence on the microstructure and tensile properties of as-built and annealing samples. The manuscript is quite well written, however, aspects of microstructure and martensitic transformation during SLM are well known and described in the literature. Also, the process parameters are common, especially the laser power. Below, there are detailed comments that have to be answered and corrected prior to publishing.
Detailed comments:
1) What is the total porosity and oxygen content of the as-built specimens?
2) How did you measure the E modulus? Was an extensometer during tensile testing used?
3) What do you mean thermo-hyteresis? Is it thermal hysteresis?
4) Fig. 3b can be removed.
5) Each parameter: UTS, YS, E and HV should have an average value and standard deviation.
6) The plasticity of stress-relieving specimens significantly improved (from 2.9% to 5%), however this is far below 10% according to the requirements of the ASTM F 3001-14 and ISO 5832 standards. The ductility problem of the SLM samples obtained at this work should be discussed.
7) Quantitative phase composition (in %) of both as-built and stress relieving samples should be calculated and presented.
8) Point in Fig. 6 where the twin lamellar glutted with dislocation lines are.
9) The TEM images of the stress-relieving sample should also be presented and compared with TEM microstructure of as-built material.
10) There is no explanation about the differences in the [1012] and [1121] twinning planes reported elsewhere [36, 37] with the [1011] twinning plane reported by the Authors.
11) The first conclusion does not come out from the results of the research obtained from this work. It is a general conclusion. Also, the fourth conclusion is mistaken. The a’ phase increase tensile strength and decrease ductility.
12) Line 113: The Ti6Al4V flakes were polished… Do you mean the TEM discs?
13) Line 118: … phase structure in the room temperature translates…, change for transforms?
14) There are also some typewriting and/or grammar mistakes that should be corrected.

Author Response
Reviewer 2#
Thank you for the questions, suggestions and your teachings, especially for put up some words and grammar mistake carefully. Every comment has been considered in detail. (The SLM printer used in this experiment is commercial printer, therefore, the maturity process parameters are common for commercial application.)
Comment 1: what is the total porosity and oxygen content of the as-built specimens?
Response: Thank you for the suggestion. The total porosity has been measured at an average of 0.51% by Archimedes method as shown in the manuscript of 98-99 lines. For the 10 days revision time, the nitrogen/oxygen analyzer was not searched out, and the oxygen content of as-built samples was not measured. But we got the chemical composition of Ti6Al4V powder in which the oxygen content is lower than 0.1% as shown Table 2 in the article and the oxygen content of SLM printer during work was controlled under 0.3% in the argon atmosphere.
Comment 2: How did you measure the E modulus? Was an extensometer during tensile testing used?
Response: Thanks for the referee’s question. The E modulus was obtained directly from the Instron 8827 tensile system. And the extensometer was used during tensile test.
Comment 3 : What do you mean thermo-hyteresis? Is it thermal hysteresis?
Response: Thanks for the referee’s kind advice. It is thermal hysteresis.
Comment 4: Fig. 3b can be removed.
Response: Thanks for the referee’s kind advice. The Fig. 3(b) has been removed.
Comment 5: Each parameter: UTS, YS, E, and HV should have an average value and standard deviation.
Response: Thanks for the referee’s kind advice. The revised parameters are shown in Table 4 in revised manuscript.
Table 4 Tensile properties of SLM Ti6Al4V samples
E (GPa) | YS (MPa) | UTS (MPa) | FS (MPa) | ε fracture (%) | H (HV) | |
As-built | 107±4 | 1142±17 | 1235±37 | 1235±37 | 3.4±1.2 | 395±21 |
Stress relieving | 114±2 | 1057±25 | 1130±30 | 1129±30 | 8.3±1.6 | 390±18 |
Comment 6: The plasticity of stress-relieving specimens significantly improved (from 2.9% to 5%), however this is far below 10% according to the requirements of ASTM F 3001-14 and ISO 5832 standards. The ductility problem of the SLM samples obtained at this work should be discussed.
Response: Thanks for the referee’s suggestion. I have read the ASTM F 3001-14 and ISO 5832 standards and revised the description about the plasticity. The ductility problem also has been discussed in revised manuscript. There are three possible factors that cause the poor plasticity compared with other reports. 1: unlike other literature where the print parameters are same in total sample, the present experiment has two sets of parameters applied in center area and edge area respectively. The brittle edge area was designed to remove easily by post-machining for commercial application, which is probably the main reason results in the samples yield early, and advanced fracture during tensile test. 2: though the support structure area was mechanically polished, there are still pits and bumps remained in such a brittle sample surface, which may accelerates the fracture process. 3: as shown in Fig. 6 as-built images, so many high density of dislocation and twins, as, the larger thermal stresses typically occurred during SLM, and that led to the error come out in tensile test due to the samples a little curled.
Comment 7: Quantitative phase composition (in%) of both as-built and stress relieving samples should be calculated and presented.
Response: Thanks for the referee’s kind suggestion. The phase composition of as-built and stress relieving samples was analyzed using the software Image-Pro Plus, wherein three OM images at different position were measured respectively as shown in Table 3.
Table 3 The phase composition of as-built and stress relieving samples.
Phase Samples | α'/% | α/% | β/% |
As-built | 28.50 | 71.50 | 0 |
Stress relieving | 9.90 | 79.55 | 10.55 |
Comment 8: Point in Fig.6 where the twin lamellar glutted with dislocation lines are.
Response: Thanks for the referee’s concern. The mistake has been revised.
Comment 9:The TEM images of the stress-relieving sample should also be presented and compared with TEM microstructure of as-built material.
Response: Thank you for kind advice. The TEM images of stress relieved sample have been displayed in Fig. 6(g)(h)(i).
Comment 10: There is no explanation about the differences in the [1012] and [1121] twinning planes reported elsewhere [36,37] with the [1011] twinning plane reported by the Authors.
Response: Thanks for the referee’s suggestion. {1012} and {1121} have been observed as extension twins. The {1011} twinning plane are in HCP materials is considered as contraction twins. For the limited revision time, we are sorry for no more time to consult some reference about the detailed distinction among the three twins
Comment 11: The first conclusion does not come out from the results of the research obtained from this work. It is a general conclusion. Also, the fourth conclusion is mistaken. The α' phase increase tensile strength and decrease ductility.
Response: Thanks for the referee’s concern. Here are the revised first and 4th conclusions.
(1) After stress relieving, β phase has been observed in XRD pattern, OM metallograph and TEM images, and the composition of α' phase reduced, indicating the α' phase decomposed during annealing treatment.
(4) α' phase results in the increase in tensile strength and hardness and decrease in plasticity. The poor plasticity was ascribed to the print parameters, remained support structures and large thermal stresses.
Comment 12: Line 113: The Ti6Al4V flakes were polished…Do you mean the TEM discs?
Response: Thanks for the referee’s concern. It is TEM discs, and we have revised the description in “The prepared Ti6Al4V flakes for TEM… ”.
Comment 13: Line 118:…phase structure in the room temperature translate…” change for “transforms?
Response: The “translates” has been revised in “transforms”.
Comment 14: There are also some typewriting and /or grammar mistakes that should be corrected.
Response: Thanks for the referee’s suggestion. We have checked the whole article to avoid the typewriting and/or grammar mistakes, and we are ready to use a professional English editing service if necessary.

Reviewer 3 Report
There are many papers on the microstructure and mechanical properties of Ti6Al4V.processed by selective laser melting. The authors need to strength the novelty and significance of their work with experimental results.
Th authors need to describe the parameters of selective laser melting affecting the mechanical performance of titanium alloy and their results in conclusion and the mani contents of the manuscript.
The authors describe the analysis or add more comments on the slective laser melted structure of martensite(martensitic size) vs. mechanical properties (yield strength, ductility, etc...)
It would be better to graphical processing stages for microstructural formation if possible.
how the authors define Elastici modulus, ultimate tensile strength and fracture strength and elongation(ductility) in the tensile test described in the manuscript (Fig.7)?
Is there is no relationship between the changes of grain size and dislocation density by selective laser melting, and yield strength of alloys?
The authors mentioned only tensile properties and morphology of fractured surface. what about the compression behavior of slective laser metlted Ti6Al4V?
It is better to provide the mechanical properties-Elastic modulus, yield strength/strain, UTS, fracture strength/strain, ductility, etc.... in a table.
Author Response
Reviewer 3#
Comment 1: There are many papers on the microstructure and mechanical properties of Ti6Al4V. processed by selective laser melting. The authors need to strength the novelty and significance of their work with experimental results.
Response: Thank you for the good advice. The authors consider seriously about the novelty and significance of this work. Also, the parameters are common and many researches on the microstructure and mechanical properties. However, the prior parameters that the commercial additive manufacturing must use is the maturity parameters. And we tend to explain the phenomenon or tensile behavior derived from the theory, and the revised content increased the depth of the analysis in martensite and mechanical properties based on the suggestions.
The SLM printer used in this experiment is commercial SLM printer. And this commercial printed mode is differ from others results, which results in the poor plasticity in comparison with other reports. The authors analyzed the possible reasons about the phenomenon. The result may be useful for industry.
And other aspects of using Feature-Scan to analyze microstructures and mechanical properties, a comparison between as-built TEM images and stress relieving images etc.
Comment 2: The authors need to describe the parameters of selective laser melting affecting the mechanical performance of titanium alloy and their results in conclusion and the main contents of the manuscript.
Response: Thanks for the referee’s suggestion. A brief description of the parameters used in SLM is appear in the manuscript (322 lines). And the different printing mode of SLM affecting the yield strength and plasticity has been discussed due to its difference with other report. (288-298 lines). Further, the laser power, scan speed and laser spot size etc. of this SLM printer are the optimized parameters for tooth manufacture and other applications. For the changeless parameters and without comparison, the description of these parameters are baseless if we try to explain. The main content of manuscript has been revised in the abstract.
Comment 3: The authors describe the analysis or add more comments on the selective laser melted structure of martensite(martensitic size) vs. mechanical properties (yield strength, ductility, etc...)
Response: Thanks for the referee’s suggestion. The more comments about martensitic size and composition, yield strength and plasticity have been supplemented in the article (188, 192, 230, 245, 275, 294,326 lines).
Comment 4: It would be better to graphical processing stages for microstructural formation if possible.
Response: Thanks for the referee’s kind advice. We are very interested in the graphical processing stage of microstructure formation. At first, we considered about simulating the temperature field during SLM by Ansys, however, the process can’t be completed in the limited revision time. Then, we thought about the crystal structure changing during SLM, for instance, if we can simulate the martensitic transformation from β phase (BCC) to α' phase (HCP). But the martensite shear model has been questioned when we found it’s not a well-established theory. And we can achieve this martensite shear model if you consider it is feasible.
Comment 5: how the authors define Elastic modulus, ultimate tensile strength and fracture strength and elongation(ductility) in the tensile test described in the manuscript (Fig.7)?
Response: Thank you for the suggestion. I am sorry for there is no detail tensile data provided before, and the Table 4 in the new manuscript displays the detailed data of elastic modulus, yield strength, ultimate tensile strength, fracture strength, elongation and hardness. And the explanation is mentioned in the 234, 264 lines.
Elastic modulus is the ratio coefficient of stress and strain during elastic deformation stage. Ultimate tensile strength is the maximum value of tensile strength, fracture strength is the tensile strength at fracture. In this study, the UTS and YS have the same value in as-built samples while the value of UTS is a little higher than YS in stress relieved samples. Fracture elongation is also measured to get the percentage of length after fracture and original length.
Comment 6: Is there is no relationship between the changes of grain size and dislocation density by selective laser melting, and yield strength of alloys?
Response: Thanks for the referee’s kind advice. The relationship between grain size/composition and dislocation density/twins, and YS/UTS has been discussed in the article. Here are a brief summary.
The refined grains will enhance the plasticity of the stress relieved samples, and then reduce the YS and UTS. The result is coincide considerably with the tensile data in the Table 4. And the twins and high dislocation density indicate that the high inner stresses certainly exist, which results in the improvement of yield strength (YS) and ultimate tensile strength (UTS).
Comment 7: The authors mentioned only tensile properties and morphology of fractured surface. what about the compression behavior of selective laser melted Ti6Al4V?
Response: Thanks for the referee’s suggestion. There are many literatures about tensile test and compression behavior of SLM Ti6Al4V, therefore, the authors select tensile experiments in order to elucidate the relationship between microstructure (α' phase, α phase and β phase) and mechanical behavior in the present study. Additionally, the authors’ following experiments will surrounding the corrosion behavior of SLM manufactured Ti6Al4V braces during in vitro oral environments, and the compression behavior has been designed in that experiment.
Comment 8: It is better to provide the mechanical properties-Elastic modulus, yield strength/strain, UTS, fracture strength/strain, ductility, etc.... in a table.
Response: Thanks for the referee’s kind advice. The data of mechanical properties are listed in Table 4, and the yield strain and fracture strain are mentioned in the article at 261-262 lines.

Round 2
Reviewer 2 Report
The manuscript was revised thoroughly, indeed, however, there are still some mistakes and sentences that have to be analyzed and revised.
First of all there are some typewriting mistakes (e.g. line 196, line 267, line 330). Also, I suggest to enlarge the scale bars in Fig. 6. There is a comment to the data presented in Table 4 (line 303): the ε fracture (%) values for both samples are not consistent with the results presented in Fig. 7 and in the text, correct it. The sentence on lines 339-340: “The refined grains will enhance the plasticity of the stress relieved samples, and then reduce the YS and UTS” seems to not be true, because the grain refinement usually increase yield stress of materials (Hall-Petch strengthening mechanism). In my opinion the effect of the plasticity improvement is more complex, since the annealing process changed the fracture mechanism of the sample.
Final suggestion, read the article again and correct remarks listed above.

Author Response
Thanks again for your suggestions.
Comment 1: First of all there are some typewriting mistakes (e.g. line 196, line 267, line 330).
Response: The words and sentences have been corrected (line 197, line 267, line 329).
Comment 2: Also, I suggest to enlarge the scale bars in Fig. 6.
Response: The scale bars has been revised.
Comment 3: There is a comment to the data presented in Table 4 (line 303): the ε fracture (%) values for both samples are not consistent with the results presented in Fig. 7 and in the text, correct it.
Response: The ε fracture(%) values has been recalculated.
Comment 4: The sentence on lines 339-340: “The refined grains will enhance the plasticity of the stress relieved samples, and then reduce the YS and UTS” seems to not be true, because the grain refinement usually increase yield stress of materials (Hall-Petch strengthening mechanism). In my opinion the effect of the plasticity improvement is more complex, since the annealing process changed the fracture mechanism of the sample.
Response: Thanks for your reminder, we have reconsidered the inexact conclusion and the details are shown in line 338 (red words).
The article has been read again and we have contacted a native English speaker to check this article.

Reviewer 3 Report
I received the author's response and the manuscript has overall quality of MDPI-Materials publication.
Author Response
Thank you very much.